# Optimal Input Gain: All You Need to Supercharge a Feed-Forward Neural Network

## Abstract

Deep learning training training algorithms are a huge success in recent years in many fields including speech, text,image video etc. Deeper and deeper layers are proposed with huge success with resnet structures having around 152 layers. Shallow convolution neural networks(CNN's) are still an active research, where some phenomena are still unexplained. Activation functions used in the network are of utmost importance, as they provide non linearity to the networks. ReLU's are the most commonly used activation function.We show a complex piece-wise linear(PWL) activation in the hidden layer. We show that these PWL activations work much better than ReLU activations in our networks for convolution neural networks and multilayer perceptrons. Result comparison in PyTorch for shallow and deep CNNs are given to further strengthen our case.

## 1 Introduction

Multilayer perceptron (MLP) neural networks are used to solve a variety of real-life approximation tasks, including stock market time series forecastingWhite (1988), power load forecastingKe et al. (2019), prognostics Kara (2021), well log processing Wang et al. (2020), currency exchange rate predictionChen et al. (2021a), control applicationsLewis et al. (1997) and stock and weather predictionMeesomsarn et al. (2009)Bochenek & Ustrnul (2022). MLPs are also used in classification problems such as speech recognitionAdolfi et al. (2023), fingerprint recognitionWang et al. (2022), character recognitionChen et al. (2021b), and face detectionKasar et al. (2016). In recent times, they also form the back end of deep learning architectures as Tyagi (2018), Qi et al. (2017).

The "no free lunch" theorem (NFL) Duda et al. (2012),Wolpert (1996) implies that no single discriminant training algorithm is universally superior. Despite this, feed-forward neural nets, or MLPs, have gained increasing popularity for two reasons. First, MLPs have the ability to approximate any continuous discriminant function with arbitrary accuracy due to universal approximation capability Girosi & Poggio (1989), Cybenko (1989), White (1990), Hartman et al. (1990) meaning that it can approximate the best classifier. However, a feed-forward network with a single layer may struggle to learn and generalize correctly due to insufficient learning, a lack of deterministic relationship between inputs and outputs, or an insufficient number of hidden units Pao (1989), Werbos (1974), Goodfellow et al. (2016). Second, with proper training, an MLP can approximate the Bayes discriminant Suter (1990) or the minimum mean-square error (MMSE) estimator Geman et al. (1992),Manry et al. (1996),Wu (1996).

Training an MLP is an unconstrained optimization problem that usually involves first order gradient methods such as backpropagation (BP), scaled conjugate gradient (SCG) Fitch et al. (1991)Møller (1993), OWO-BP Hecht-Nielsen (1992), Tyagi et al. (2022a) and second order Levenberg-Marquardt (LM) Battiti (1992)Hagan & Menhaj (1994) and Newton's algorithm Tyagi et al. (2021), Tyagi et al. (2021) and OWO-MOLF Tyagi et al. (2020). Within first and second order, training algorithms can be one stage, in which all the weights of the network are updated simultaneously and two stage, in which input and output weights are trained alternately Tyagi (2018). However, each of these approaches has its limitations. Newton's and LM scale worse than OWO-BP. OWO-BP takes $O(N_w^2)$ operations for sufficiently large $N_w$, where $N_w$ is the total weights of the network. It is also unduly slow in a flat error surface and could be a more reliable learning paradigm.

OWO-BP also lacks affine invariance Tyagi et al. (2022a). SCG scales well but has no internal mechanism to overcome linearly dependent inputs Tyagi (2018). Newton's method is a second-order algorithm that requires computing and storing the Hessian matrix at each iteration, which can be computationally expensive Tan & Lim (2019). The LM algorithm is faster than Newton's method because it approximates the Hessian matrix using the Jacobian matrix Levenberg (1944). However, it becomes less efficient as the number of parameters increases Tan & Lim (2019).

MLP training is also sensitive to many parameters of the network and its training data, including the input means LeCun et al. (1998a), the initial network weights Rumelhart et al. (1986), Kolen & Pollack (1990), and sensitive to the collinearity of its inputs Hashem & Schmeiser (1995). Also, scaling the network architecture to learn more complex representations is cumbersome. This limited applications involving MLP to challenging but less complex problems where shallow architectures could be used to learn and model the behavior effectively. The recent developments of transformers Vaswani et al. (2017) and their success in complex applications involving natural speech Dong et al. (2018), Pham et al. (2019) and vision Kolesnikov et al. (2021) have renewed interests in feed-forward network architectures, as they form the building blocks to the more complex transformer architectures int (2023). Feed-forward network are also being used in active production for radar perception stack in autonomous driving. Tyagi et al. (2022b).

We present a family of fast learning algorithms targeted towards training a fixed architecture, fully connected multi-layer perceptron with a single hidden layer capable of learning from both approximation and classification datasets. In Malalur & Manry (2009), a method for optimizing input gains called the optimal input gain (OIG) was presented. Preliminary experiments showed that when this method was applied to the first-order MLP training method like the BP, it significantly improved the overall network's performance with minimal computational overhead. However, the method performs less optimally under the presence of linearly dependent inputs. In general, this is true for other MLP training algorithms as well. We expand the idea of OIG to apply another first-order two-stage training algorithm called hidden weight optimization (HWO) Yu et al. (2004), Tyagi et al. (2022a) to formulate the OIG-HWO algorithm.

Following Malalur & Manry (2009), we expand on the details of the motivation behind the OIG algorithm, along with a thorough analysis of its structure, performance, and limitation. In addition, we propose an improvement to overcome the limitation and compare the new algorithm's performance with existing algorithms. Our vision for the OIG-HWO presented in this paper is twofold, firstly, to be a strong candidate for challenging but less complex applications that can rival available shallow learning architectures in speed and performance, and secondly, to serve as a potential building block for more complex deep learning architectures.

The rest of the paper is organized as follows. Section II covers the basics of MLP notation and training and an overview of existing algorithms. Section III discusses the linear transformation of inputs. In Section IV, we describe the OIG-HWO algorithm and its training process. Finally, in Section V, we compare the results of the OIG-HWO algorithm to those obtained using existing approaches on approximation data and classification data used for replacement in deep learning classifiers.

## 2 Prior work

### 2.1 Structure and notation

A fully connected MLP with one hidden layer is shown in Figure 1. Input weight $w(k, n)$ connects the $n^{th}$ input $x_p(n)$ to the $k^{th}$ hidden unit. Output weight $w_{oh}(i, k)$ connects the $k^{th}$ hidden unit's activation $o_p(k)$ to the $i^{th}$ output $y_p(i)$, which has a linear activation. The bypass weight $w_{oi}(i, n)$ connects $x_p(n)$ to $y_p(i)$. In the training data $\{\mathbf{x}_p, \mathbf{t}_p\}$ for a fully connected MLP, the $p^{th}$ input vector $\mathbf{x}_p$ is initially of dimension N and the $p^{th}$ desired output (target) vector $\mathbf{t}_p$ has dimension M. The pattern number p varies from 1 to $N_v$. Let the input vectors be augmented by an extra element $x_p(N + 1)$ where $x_p(N + 1) = 1$, so $\mathbf{x}_p = [x_p(1), x_p(2)\dots x_p(N + 1)]^T$. Weights leading away from $x_p(N + 1)$ are hidden or output layer thresholds. For the $p^{th}$ pattern, the hidden unit's output vector $\mathbf{n_p}$ can be written as

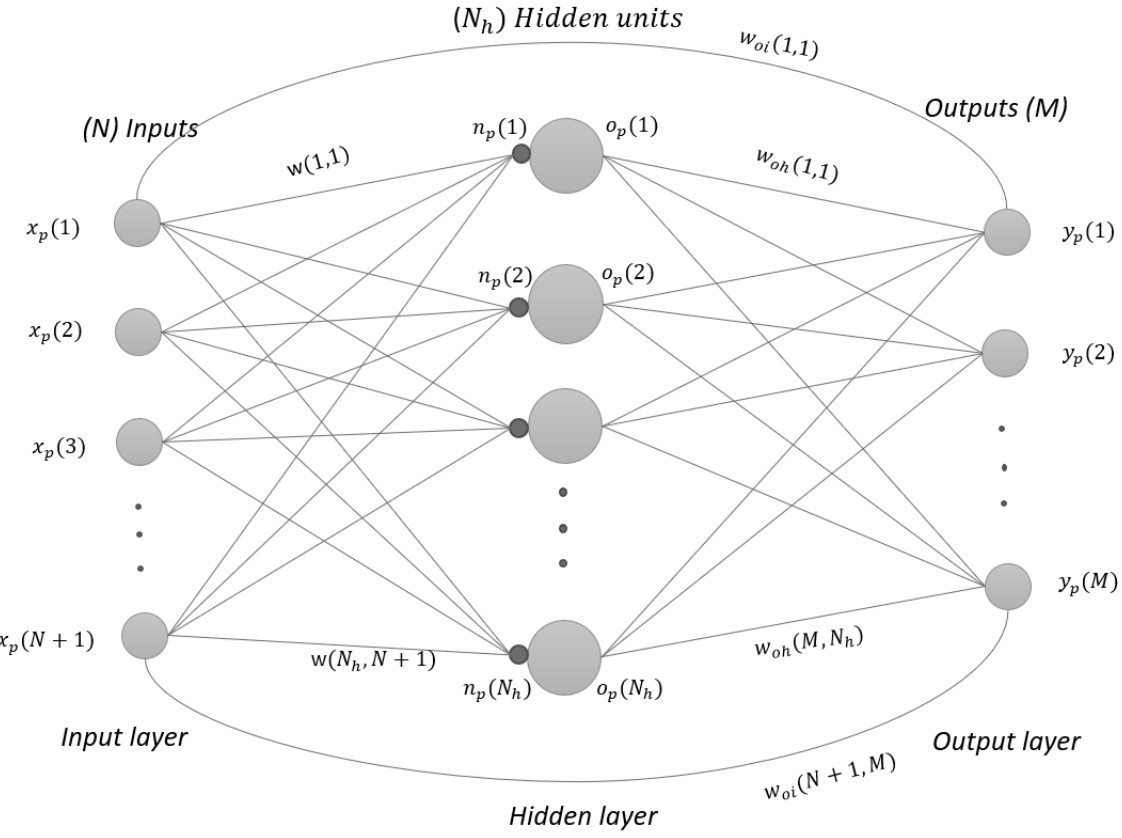

Figure 1: Single hidden layer fully connected MLP

$$\mathbf{n}_p = \mathbf{W} \cdot \mathbf{x}_p \tag{1}$$

where $\mathbf{n}_p$ is of size $N_h$ by 1, and the input weight matrix $\mathbf{W}$ is of size $N_h$ by $(N + 1)$. For the $p^{th}$ pattern, the $k^{th}$ hidden unit's output, $o_p(k)$, is calculated as $o_p(k) = f(n_p(k))$, where $f(.)$ denotes a nonlinear hidden layer activation function such as the rectified linear unit ($ReLU$) which is given as follows Nair & Hinton (2010).

$$f(n_p(k)) = max(0, n_p(k)) = \begin{cases} n_p(k), & if \quad n_p(k) \geq 0 \\ 0, & if \quad n_p(k) < 0 \end{cases} \tag{2}$$

The $M$ dimensional network output vector $\mathbf{y_p}$ is

$$\mathbf{y}_p = \mathbf{W}_{oi} \cdot \mathbf{x}_p + \mathbf{W}_{oh} \cdot \mathbf{o}_p \tag{3}$$

where $\mathbf{o}_p$ is the $N_h$ dimensional hidden unit activation vector. The last columns of $\mathbf{W}$ and $\mathbf{W}_{oi}$ respectively store the hidden unit and output unit threshold values. During training the unknown weights are solved by minimizing a mean-squared error (MSE) function described as

$$E = \frac{1}{N_v} \sum_{p=1}^{N_v} \sum_{i=1}^{M} [t_p(i) - y_p(i)]^2 \tag{4}$$

Training a neural network involves formulating it as an unconstrained optimization problem and then applying a learning procedure. Typically, the learning procedure is a line search Gill et al. (2019), with a layer-by-layer

optimization Biegler-König & Bärmann (1993), Zhang et al. (1999), Wang & Chen (1996), involving first and second-order algorithms.

## 2.2 Scaled conjugate gradient algorithm

Conjugate gradient (CG) Tyagi et al. (2022a) line-searches in successive conjugate directions and has faster convergence than steepest descent. To train an MLP using the CG algorithm (CG-MLP), we update all the network weights $\mathbf{w}$ simultaneously as follows:

$$\mathbf{w} \leftarrow \mathbf{w} + z \cdot \mathbf{p} \tag{5}$$

where $z$ is the learning rate that can be derived as LeCun et al. (1998a),Tyagi et al. (2022a).

$$z = -\frac{\frac{\partial E(\mathbf{w}+z \cdot \mathbf{p})}{\partial z}}{\frac{\partial^2 E(\mathbf{w}+z \cdot \mathbf{p})}{\partial z^2}}|_{z=0} \tag{6}$$

The direction vector $\mathbf{p}$ is obtained from the gradient $\mathbf{g}$ as

$$\mathbf{p} \leftarrow -\mathbf{g} + B_1 \cdot \mathbf{p} \tag{7}$$

where $\mathbf{p} = vec\,(\mathbf{P}, \mathbf{P}_{\text{oh}}, \mathbf{P}_{\text{oi}})$ and $\mathbf{P}$, $\mathbf{P}_{\text{oh}}$ and $\mathbf{P}_{\text{oi}}$ are the direction vectors corresponding to weight arrays $(\mathbf{W}, \mathbf{W}_{\text{oh}}, \mathbf{W}_{\text{oi}})$. CG uses backpropagation to calculate $\mathbf{g}$. $B_1$ is the ratio of the gradient energy from two consecutive iterations. If the error function were quadratic, CG would converge in $N_w$ iterations Boyd & Vandenberghe (2004), where the number of network weights is $N_w = dim(\mathbf{w})$. CG is scalable and widely used in training large datasets, as the network Hessian is not calculated Le et al. (2011). Therefore, in a CG, the step size is determined using a line search along the direction of the conjugate gradient.

SCG Møller (1993) scales the conjugate gradient direction by a scaling factor determined using a quasi-Newton approximation of the Hessian matrix. This scaling factor helps to accelerate the algorithm's convergence, especially for problems where the condition number of the Hessian matrix is large. SCG requires the computation of the Hessian matrix (or an approximation) and its inverse. Other variations of CG exist Tyagi et al. (2014). However, in this study, we choose to use SCG.

## 2.3 Levenberg-Marquardt algorithm

The Levenberg-Marquardt (LM) algorithm Tyagi et al. (2022a) is a hybrid first- and second-order training method that combines the fast convergence of the steepest descent method with the precise optimization of the Newton method Levenberg (1944). However, inverting the Hessian matrix $\mathbf{H}$ can be challenging due to its potential singularity or ill-conditioning Bishop (2006). To address this issue, the LM algorithm introduces a damping parameter $\lambda$ to the diagonal of the Hessian matrix as

$$\mathbf{H}_{LM} = \mathbf{H} + \lambda \cdot \mathbf{I} \tag{8}$$

where $\mathbf{I}$ is an identity matrix with dimensions equal to those of $\mathbf{H}$. The resulting matrix $\mathbf{H}_{LM}$ is then nonsingular, and the direction vector $\mathbf{d}_{LM}$ can be calculated by solving:

$$\mathbf{H}_{LM}\mathbf{d}_{LM} = \mathbf{g} \tag{9}$$

The constant $\lambda$ represents a trade-off value between first and second order for the LM algorithm. When $\lambda$ is close to zero, LM approximates Newton's method and has minimal impact on the Hessian matrix. When $\lambda$ is large, LM approaches the steepest descent and the Hessian matrix approximates an identity matrix. However, the disadvantage of the LM algorithm is that it scales poorly and is only suitable for small data sets Tyagi et al. (2022a).

## 2.4  Output weight optimization

Output weight optimization (OWO) Barton (1991), Tyagi et al. (2021) is a technique to solve for $W_{oh}$ and $W_{oi}$ . Equation (3) can be re-written as

$$\mathbf{y}_p = \mathbf{W}_o \cdot \mathbf{x}_{ap} \tag{10}$$

where $\mathbf{x}_{ap} = [\mathbf{x}_p^T : \mathbf{o}_p^T]^T$ is the augmented input column vector of size $N_u = N + N_h + 1$. $\mathbf{W}_o$ is formed as $[\mathbf{W}_{oi} : \mathbf{W}_{oh}]$ of dimensions $M$ by $N_u$. The output weights can be found by setting $\partial E / \partial W_o = 0$, which leads to the set of linear equations

$$\mathbf{C} = \mathbf{R} \cdot \mathbf{W}_o^T \tag{11}$$

where $\mathbf{C} = \frac{1}{N_v} \sum_{p=1}^{N_v} \mathbf{x}_{ap} \mathbf{t}_p^T$ and $\mathbf{R} = \frac{1}{N_v} \sum_{p=1}^{N_v} \mathbf{x}_{ap} \mathbf{x}_{ap}^T$. Equation (11) can be solved using orthogonal least squares (OLS) methods Tyagi (2018). OWO provides fast training and avoids local minima Manry et al. (1994). However, it only trains the output weights.

## 2.5  Input weight optimization

Input weight optimization Tyagi (2018) is a technique for iteratively improving $\mathbf{W}$ via steepest descent. The $N_h$ by $(N + 1)$ negative input weight Jacobian matrix for the $p^{th}$ pattern's input weights is

$$\mathbf{G} = \frac{1}{N_v} \sum_{p=1}^{N_v} \delta_p \mathbf{x}_p^T \tag{12}$$

where $\boldsymbol{\delta_p} = [\delta_p(1), \delta_p(2), ...., \delta_p(N_h)]^T$ is the $N_h$ by 1 column vector of hidden unit delta functions Rumelhart et al. (1985). $\mathbf{W}$ is updated in a given iteration as

$$\mathbf{W} = \mathbf{W} + z \cdot \mathbf{G} \tag{13}$$

where $z$ is the learning factor. Combined with BP, we formulate OWO-BP Tyagi et al. (2022a), a two-stage training algorithm developed as an alternative to BP. In a given iteration of OWO-BP, we first find the weights, $W_{oh}$ and $W_{oi}$ and then separately train $\mathbf{W}$ using BP Tyagi (2018). OWO-BP is attractive for several reasons. First, the training is faster since solving linear equations for output weights in a given iteration is faster than using a gradient method. Second, when OWO optimizes output weights for a given input weight matrix, some local minima are avoided. Third, the method exhibits improved training performance compared to using only BP to update all the weights in the network. It can be shown that OWO-BP converges, and it leads to the convergence of the weights to a critical point in weight space Tyagi et al. (2022a). This can be a global minimum, a local minimum, or a saddle point.

## 2.6  Hidden weight optimization

HWO Scalero & Tepedelenlioglu (1992), finds an improved gradient matrix $\mathbf{G_{hwo}}$ by solving the following linear equation

$$\mathbf{G}_{hwo} \cdot \mathbf{R_i} = \mathbf{G} \tag{14}$$

where $\mathbf{R_i}$ is the input autocorrelation matrix as

$$\mathbf{R_i} = \frac{1}{N_v} \sum_{p=1}^{N_v} \mathbf{x}_p \mathbf{x}_p^T \tag{15}$$

and $\mathbf{G}$ is the backpropagation negative gradient matrix Rumelhart et al. (1985). Equation (14) can be rewritten as

$$\mathbf{G}_{hwo} = \mathbf{G} \cdot \mathbf{R_i}^{-1} \tag{16}$$

where $\mathbf{G}_{hwo} = \mathbf{G} \cdot \mathbf{A}^T \cdot \mathbf{A}$. Equation (14) can be solved using OLS or matrix inversion using the singular value decomposition (SVD). $\mathbf{A}$ is the whitening transform matrix Raudys (2001), Tyagi et al. (2020). It is shown in Robinson & Manry (2013) that HWO is equivalent to applying a whitening transform to the training data to de-correlate it. $\mathbf{W}$ is now updated using $\mathbf{G_{hwo}}$ instead of $\mathbf{G}$ as

$$\mathbf{W} = \mathbf{W} + z \cdot \mathbf{G}_{hwo} \tag{17}$$

## 3 Proposed work

The study of the effects of applying the equivalent networks theory to the augmented input vectors $\mathbf{x}_p$ is thoroughly discussed in Malalur & Manry (2009). In this work, we build upon the concept presented in Malalur & Manry (2009), Nguyen et al. (2016) and examine the impact of transformed input gains on the training process in conjunction with HWO.

### 3.1 Mathematical background

Consider a nonlinear network designated as *MLP-1* with inputs $\mathbf{x} \in \mathbb{R}^{N+1}$, where the restriction $\mathbf{x}_{N+1} = 1$ is imposed, and outputs $\mathbf{y} \in \mathbb{R}^M$. Another network, referred to as *MLP-2*, has inputs $\mathbf{x}' = \mathbf{A} \cdot \mathbf{x}$ and outputs $\mathbf{y}' \in \mathbb{R}^M$. These two networks are considered strongly equivalent if, for all $\mathbf{x}_p \in \mathbb{R}^{N+1}$, we have $\mathbf{y}'_p = \mathbf{y}_p$. The network *MLP-1* is trained on the original input vectors $\mathbf{x}_p$, while *MLP-2* is trained using the transformed input vectors $\mathbf{x}'_p$ defined as

$$\mathbf{x}'_p = \mathbf{A} \cdot \mathbf{x}_p \tag{18}$$

where $\mathbf{A}$ is an $N'$ by $(N+1)$ rectangular transformation matrix, for some $N' \geq (N+1)$. We establish in Malalur & Manry (2009) that input weights for *MLP-1* and *MLP-2* are related as

$$\mathbf{W}' \cdot \mathbf{A} = \mathbf{W} \tag{19}$$

The negative gradient matrix for training the input weights in *MLP-2* is given by $\mathbf{G}' = \mathbf{G} \cdot \mathbf{A}^T$. Now, suppose that this negative gradient $\mathbf{G}'$ for *MLP-2* is mapped back to modify the input weights in *MLP-1*, using equation (19). The resulting mapped negative gradient for *MLP-1* is then

$$\begin{aligned} \mathbf{G}'' &= \mathbf{G} \cdot \mathbf{R}_i \\ \mathbf{R}_i &= \mathbf{A}^T \cdot \mathbf{A} \end{aligned} \tag{20}$$

By expressing the SVD of $\mathbf{R_i}$ as $\mathbf{R_i} = \mathbf{U\Sigma U}^T$, we can derive that $\mathbf{R_i}^{-1} = \mathbf{U\Sigma}^{-1}\mathbf{U}^T$, where $\mathbf{U}$ is an orthogonal matrix, $\mathbf{\Sigma}$ is a diagonal matrix with the singular values of $\mathbf{R_i}$, and $\mathbf{\Sigma}^{-1}$ is the diagonal matrix with the reciprocal of the non-zero singular values of $\mathbf{R_i}$. Using equation (20), it can be deduced that $\mathbf{A} = \mathbf{\Sigma}^{\frac{1}{2}}\mathbf{U}^T$. Comparing equation (20) with equation (16), it is clear that performing OWO-HWO is equivalent to performing OWO-BP on input vectors to which the whitening transformation Robinson & Manry (2013) has been applied. Since BP with optimal learning factor (OLF) converges, it is clear that HWO with an OLF also converges.

***Lemma-1***: *If we are at a local minimum in the weight space of the original network, we are also at a local minimum in the weight space of the transformed network.*

This follows from ((20) if G = 0.

***Lemma-2***: *If the input weight matrix $\mathbf{W}'$ of the transformed network is trained using BP, this is not equivalent to applying BP to the original network's weight matrix $\mathbf{W}$ unless the matrix $\mathbf{A}$ is orthogonal.*

This can be derived from equation (20) because for any orthogonal matrix $\mathbf{A}$, equation (20) becomes $\mathbf{G}'' = \mathbf{G}$. This is also intuitive if we consider that BP updates the weights in the direction of the negative gradient of the loss function with respect to the weights.

***Lemma-3***: *For a non-diagonal matrix $\mathbf{R}$, there exist an uncountably infinite number of matrices $\mathbf{A}$ that can be constructed.*

This follows from (20). It is because a non-diagonal matrix $\mathbf{R}$ has at least one non-zero element not located on the main diagonal. As there are infinite choices for the values of these non-zero elements, there are an uncountable number of possible matrices $\mathbf{A}$ that can be constructed by choosing the values of these non-zero elements in $\mathbf{R}$.

If the transformation matrix $\mathbf{A}$ is not orthogonal, then the mapped negative gradient for *MLP-1* obtained from *MLP-2* will not be equivalent to the true negative gradient of the loss function with respect to the weights in *MLP-1*. As a result, when optimal learning factors are used with BP to train the input weights, training with the original data is equivalent to training with the transformed data. Therefore, orthogonal transform matrices are useless in this context, as mentioned in Yu et al. (2005). Using the results derived in this section, many ways to improve feed-forward training algorithms suggest themselves. The intuition behind the proposed work is based on the following three ideas :

1. Choose a training algorithm that utilizes the negative gradient matrix $\mathbf{G}$.

2. Substitute the negative gradient matrix $\mathbf{G}$ with the modified matrix $\mathbf{G}''$ from equation (20).

3. Identify appropriate elements in the matrix $\mathbf{R}$.

Single-stage optimization algorithms, such as conjugate gradient (CG) Tyagi et al. (2022a), may be suitable for addressing this problem. However, incorporating the elements of $\mathbf{R}$ as additional weights in the optimization process may compromise the conjugacy of the direction vectors if $\mathbf{R}$ is solved for at each iteration. As an alternative, using two-stage training algorithms that utilize the negative gradient matrix $\mathbf{G}$ or direction matrix $\mathbf{D}$, such as OWO-BP and OWO-HWO Chen et al. (1999). In this work, we focus on OIG to develop OIG-HWO. Specifically, we will develop a method for solving the matrix $\mathbf{R}$, compute the resulting Gauss-Newton approximate Hessian for $\mathbf{R}$, and apply the resulting OIG-HWO to improve the performance of OWO-BP.

### 3.2 Optimal Input Gain algorithm

There are at least two intuitive approaches for optimizing input gains to improve the performance of a given training algorithm. To minimize the training error $E$, these approaches involve searching for either the matrix $\mathbf{A}$ or the resulting matrix $\mathbf{R}$ in each iteration to minimize the training error $E$. As stated in *Lemma-2*, optimizing $\mathbf{R}$ will likely yield fewer solutions. In this section, we describe a method for solving $\mathbf{R}$, find the resulting Gauss-Newton approximate Hessian for $\mathbf{R}$, and use the resulting OIG algorithm to improve OWO-BP. The simplest non-orthogonal, non-singular transform matrix $\mathbf{A}$ is diagonal. For this case, let $r(k)$ initially denote the $k^{th}$ diagonal element of $\mathbf{R}$. Also, the elements of $\mathbf{x}'_p$ are simply scaled versions of $\mathbf{x}_p$. Following

(20) we get

$$\boldsymbol{R} = \begin{bmatrix} r(1) & 0 & \cdots & 0 & 0 \\ 0 & r(2) & \cdots & 0 & 0 \\ \vdots & \vdots & \ddots & \vdots & \vdots \\ 0 & 0 & \cdots & r(N) & 0 \\ 0 & 0 & \cdots & 0 & r(N+1) \end{bmatrix} \tag{21}$$

Instead of using only the negative gradient elements $g(k,n)$ to update the input weights, we use $g(k,n) \cdot r(n)$ to replace $g(k,n)$, the elements matrix $\mathbf{G}$ in equation 17. It is also noteworthy that the optimal learning factor (OLF), $z$ Tyagi et al. (2022a) be absorbed into the gains $r(n)$. Consider a multi-layer perceptron (MLP) being trained using the OWO-BP algorithm. The negative gradient $\mathbf{G}$ is a matrix of dimensions $N_h$ by $(N+1)$, and the error function to be minimized with respect to the gains $r(n)$ is given in (4). This error function is defined as follows:

$$y_p(i) = \sum_{n=1}^{N+1} w_{oi}(i,n)x_p(n) + \sum_{k=1}^{N_h} w_{oh}(i,k)\cdot$$
$$f\left(\sum_{n=1}^{N+1}(w(k,n)+r(n)\cdot g(k,n))x_p(n)\right) \tag{22}$$

The first partial of $E$ with respect to $r(m)$ is

$$d_r(m) \equiv \frac{\partial E}{\partial r(m)} = \frac{-2}{N_v} \sum_{p=1}^{N_v} x_p(m)$$

$$\sum_{i=1}^{M} [t_p(i) - y_p(i)] v(i, m) \tag{23}$$

Here, $g(k, m)$ is an element of the negative gradient matrix $\mathbf{G}$ in equation (12), and $o'_p(k)$ denotes the derivative of $o_p(k)$ with respect to its net function. Then,

$$v(i, m) = \sum_{k=1}^{N_h} w_{oh}(i, k) o'_p(k) g(k, m) \tag{24}$$

Using Gauss-Newton updates Bishop (2006), the elements of the Hessian matrix $\mathbf{H_{ig}}$ are

$$h_{ig}(m, u) \equiv \frac{\partial^2 E}{\partial r(m) \partial r(u)} = \frac{2}{N_v} \sum_{p=1}^{N_v} x_p(m) x_p(u) \cdot$$

$$\sum_{i=1}^{M} v(i, m) v(i, u) \tag{25}$$

Finally, the input gain coefficient vector $\mathbf{r}$ is calcualted using OLS by solving

$$\mathbf{H_{ig}} \cdot \mathbf{r} = \mathbf{d}_r \tag{26}$$

### 3.2.1 OIG Hessian matrix

We choose to use Hessian matrix to analyze the convergence properties of OIG-HWO. Equation (25) for the OIG-HWO Hessian can be re-written as,

$$h_{ig}(m, u) = \sum_{k=1}^{N_h} \sum_{j=1}^{N_h} \left[ \frac{2}{N_v} \sum_{p=1}^{N_v} x_p(m) x_p(u) o'_p(k) o'_p(k) \cdot \sum_{i=1}^{N+1} w_{oh}(i, k) w_{oh}(i, j) \right] g(k, m) \cdot g(j, u) \tag{27}$$

The term within the square brackets is nothing but an element from the Hessian of Newton's method for updating input weights. Hence,

$$h_{ig}(m, u) = \sum_{k=1}^{N_h} \sum_{j=1}^{N_h} \left[ \frac{\partial^2 E}{\partial w(k, m) \partial w(j, u)} \right] \cdot g(k, m) \cdot g(j, u) \tag{28}$$

For fixed $(m, u)$, the above equation can be expressed as

$$h_{ig}(m, u) = \sum_{k=1}^{N_h} g_m(k) \sum_{j=1}^{N_h} h_N^{m,u}(k, j) g_u(j)$$

$$= \mathbf{g_m^T H_N^{mu} g_u} \tag{29}$$

where, $g_m$ is the $m^{th}$ column of the negative gradient matrix $\mathbf{G}$ and $\mathbf{H}_N^{m,u}$ is the matrix formed by choosing elements from the Newton's Hessian for weights connecting inputs $(m, u)$ to all hidden units.

Equation (29) gives the expression for a single element of the OIG-HWO Hessian, which combines information from $N_h$ rows and columns of the Newton Hessian. This can be seen as compressing the original Newton

Hessian of dimensions $N_h$ by $(N+1)$ down to $(N+1)$. The OIG-HWO Hessian encodes the information from the Newton Hessian in a smaller dimension, making it less sensitive to input conditions and faster to compute. From equation (28), we see that the Hessian from Newton's method uses four indices $(j, m, u, k)$ and can be viewed as a 4-dimensional array, represented by $H_N^4 \in \mathbf{R}^{N_h \times (N+1) \times (N+1) \times N_h}$. Using this representation, we can express a 4-dimensional OIG-HWO Hessian as

$$\mathbf{H}_{ig}^4 = \mathbf{G}^T \mathbf{H}_N^4 \mathbf{G} \tag{30}$$

where $\mathbf{H}_{ig}^4$ are defined as,

$$h_{ig}^4(m, u, n, l) = \sum_{j=1}^{N_h} \sum_{k=1}^{N_h} h_N(j, m, u, k) g(j, n) g(k, l) \tag{31}$$

where $h_N(j, m, u, k)$ is an element of $\mathbf{H}_N^4$. Comparing (28) and (31), we see that $h_{ig}(m, u) = h_{ig}^4(m, u, m, u)$, i.e., the 4-dimensional $\mathbf{H}_{ig}^4$ is transformed into the 2-dimensional Hessian, $\mathbf{H}_{ig}$, by setting $n = m$ and $l = u$. To make this idea clear, consider a matrix, $\mathbf{Q}$, then $p(n) = q(n, n)$ is a vector, $\mathbf{p}$, of all diagonal elements of $\mathbf{Q}$. Similarly, the OIG-HWO Hessian $\mathbf{H}_{ig}$ is formed by a weighted combination of elements from $\mathbf{H}_N^4$.

### 3.2.2 OIG Integrated with OWO-BP

To minimize the error function $E$, given the vector of input gains $\mathbf{r}$, the gradient $\mathbf{d}_r$, and the Hessian $\mathbf{H}_{ig}$, we can utilize Newton's method. Two potential approaches can be taken in each iteration of this method, first is that we transform the gradient matrix using $\mathbf{R}$ as shown in equation (20), and second, we decompose $\mathbf{R}$ to find $\mathbf{A}$ using OLS, and then transform the input data according to equation (18) before applying OWO-BP with the optimal learning factor (OLF). While the second approach may be more accurate, it is also more computationally inefficient and, therefore, not practical, even when $\mathbf{A}$ is diagonal. Therefore, it is generally recommended to use the first approach in order to minimize the error function effectively. Hence, the OWO is replaced with OIG in the OWO-BP algorithm to form OIG-BP described in Algorithm 1.

---

**Algorithm 1** OIG-BP training algorithm

---

1: Initialize $\mathbf{W}, \mathbf{W_{oi}}, \mathbf{W_{oh}}, N_{it}$ , it$\leftarrow 0$
2: **while** it $< N_{it}$ **do**
3:     Solve (11) for all output weights.
4:     Calculate negative $\mathbf{G}$ using equation (12)
5:     **OIG step** Calculate $\mathbf{d_r}$ and hessian $\mathbf{H}_{ig}$ from (23) and (25) respectively.
6:     Solve for $\mathbf{r}$ using equation (26)
7:     Update $\mathbf{W} \leftarrow \mathbf{W} + \mathbf{r} \cdot \mathbf{G}$
8:     **OWO step** : Solve equation (11) to obtain $\mathbf{W_o}$
9:     it $\leftarrow$ it $+ 1$
10: **end while**

---

When there are no linearly dependent inputs, the OIG algorithm can find the optimal gain coefficients for each input that minimize the overall mean squared training error. However, this is only sometimes the case when there are linearly dependent inputs. In this scenario, it is straightforward to show that the input autocorrelation matrix $\mathbf{R}_{in}$ and the gradient matrix $\mathbf{G}$ have dependent columns. This leads to the OIG Hessian being *ill-conditioned* and to sub-optimal gain coefficients. This could cause OIG-BP to have sub-optimal performance and possibly poor convergence.

### 3.3 Improvement to OIG-BP

In order to overcome sub-optimal performance of OIG in the presence of linearly dependent inputs, we show the immunity of HWO to linearly dependent inputs. We analyze the effect of replacing BP used in OIG-BP with HWO and show that using HWO forces $H_{ig}$ to be singular for linearly dependent inputs, which is highly desirable in order to detect and eliminate the dependent inputs.

### 3.3.1 Effect of Linearly Dependent Inputs on HWO

If one of the inputs to the network is linearly dependent, it will cause the input auto-correlation matrix, $\mathbf{R}_i$, to be singular. This can affect the convergence of the CG algorithm, leading to poor training performance. In this case, using OLS may be useful for detecting and eliminating the linearly dependent input. To compute the orthonormal weight update matrix $\mathbf{G}_{hwo}$ using OLS, we first compute $\mathbf{G}'_{hwo}$ as

$$\mathbf{G}'_{hwo} = \mathbf{G} \cdot \mathbf{C}^T \tag{32}$$

where $\mathbf{C}$ is a lower triangular matrix of orthonormal coefficients of dimension $(N+1)$. We can then map the orthonormal weight update to the original weight update as

$$\begin{aligned} \mathbf{G}_{hwo} &= \mathbf{G}'_{hwo} \cdot \mathbf{C} \\ &= \mathbf{G} \cdot \mathbf{C}^T \cdot \mathbf{C} \end{aligned} \tag{33}$$

Assume $\mathbf{x}_p(N+2)$ was linearly dependent. This would cause the $(N+2)^{th}$ row and column of $\mathbf{R}_i$ to be linearly dependent. During OLS, a singular auto-correlation matrix transforms to the $(N+2)^{th}$ row of $\mathbf{C}$ to be zero. We replace BP in OIG-BP with HWO. The resulting OIG-HWO algorithm is described in Algorithm 2.

---

**Algorithm 2** OIG-HWO training algorithm

---

1: Initialize $\mathbf{W}, \mathbf{W_{oi}}, \mathbf{W_{oh}}, N_{it}$ , it$\leftarrow 0$
2: Calculate $\mathbf{R}_i$ using (15)
3: **while** it $< N_{it}$ **do**
4:    Calculate negative $\mathbf{G}$ using (12)
5:    **HWO step** : Calculate $\mathbf{G_{hwo}}$ using (33) to eliminate any linear dependency in the inputs.
6:    **OIG step**: Calculate $\mathbf{d_r}$ and hessian $\mathbf{H}_{ig}$ from (23) and (25) respectively.
7:    Solve for $\mathbf{r}$ using equation (26)
8:    Update $\mathbf{W} \leftarrow \mathbf{W} + \mathbf{r} \cdot \mathbf{G_{hwo}}$
9:    **OWO step** : Solve equation (11) to obtain $\mathbf{W_o}$
10:    it $\leftarrow$ it $+ 1$
11: **end while**

---

***Lemma-3***: *The $OIG - HWO$ algorithm is immune to linearly dependent inputs and will completely ignore the dependent inputs during training.*

Since the $(N+2)^{th}$ row of $\mathbf{C}$ will be zero, it follows that $\mathbf{C}^T\mathbf{C}$, which will be a square, symmetric matrix with zeros for the $(N+2)^{th}$ row and column. Further, from (33), $\mathbf{G}_{hwo}$ will have zeros for the $(N+2)^{th}$ column. The implication is that the weight update vector computed for all input weights connected to the dependent input $(N+2)$ is zero. These weights are not updated during training, effectively *freezing* them. This is highly desirable, as the dependent input does not contribute any new information. Thus, HWO-type update using OLS is perfectly capable of picking up linearly dependent inputs, leading to a robust training algorithm. This makes OIG-HWO immune to linearly dependent inputs.

To illustrate the meaning of *lemma-3*, we took a data set called *twod.tra* ipn (2022), and generated a second one by adding some dependent inputs. Networks for the two datasets were initialized with the same net function means and standard deviations. Figure 2 clearly shows that the two training error curves overlay each other, validating *lemma-3*.

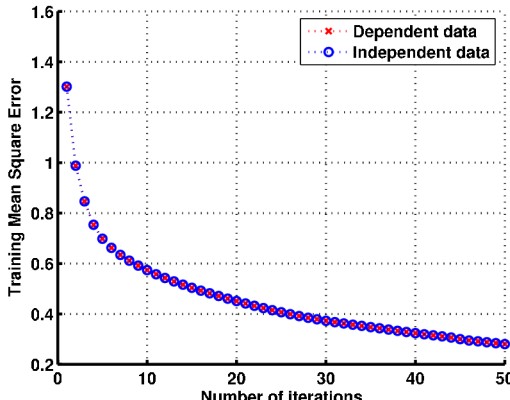

Figure 2: Immunity of OIG-HWO to linearly dependent inputs

To further demonstrate *lemma-3* and the effectiveness of OIG-HWO, we compare its performance on the dependent dataset with LM and OIG-BP. Figure 3 shows how dependence can slow down learning in all except the improved OIG-HWO algorithm. The effect is predominant in LM that takes huge computational resources.

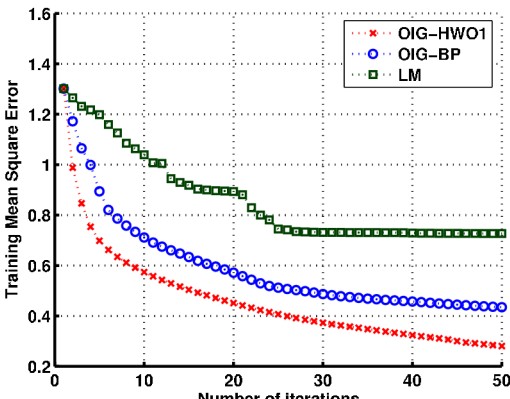

Figure 3: Performance comparison on dependent data

Mathematically, suppose that the input vector $\mathbf{x_p}$ are biased such that $E[\mathbf{x}_p] = \mathbf{m}$. A zero-mean version of $\mathbf{x_p}$ is $\mathbf{x}'_p$ which satisfies $\mathbf{x}_p = \mathbf{x}'_p + \mathbf{m}$. It is shown in Malalur & Manry (2009) that networks train more effectively with bunbiased inputs. Now, $\mathbf{x}'_p$ can be expressed as $\mathbf{A} \cdot \mathbf{x}_p$, where

$$\mathbf{A} = \begin{bmatrix} 1 & 0 & \cdots & 0 & -m_1 \\ 0 & 1 & \cdots & 0 & -m_2 \\ \vdots & \vdots & \ddots & \vdots & \vdots \\ 0 & 0 & \cdots & 1 & -m_N \\ 0 & 0 & \cdots & 0 & 1 \end{bmatrix} \tag{34}$$

Figure 3 shows that non-orthogonal transform matrices improve the training since it makes the inputs zero-mean. The HWO component of the OIG-HWO algorithm addresses the issue of sub-optimal performance in the presence of linearly dependent inputs. It has been demonstrated that the HWO is immune to such inputs Malalur & Manry (2009). By replacing the BP component of the OIG-BP with HWO, we can analyze

the effect on the singularity of $\mathbf{H}_{ig}$ for linearly dependent inputs. This is beneficial because the singularity of $\mathbf{H}_{ig}$ allows for the detection and removal of dependent inputs.

# 4 Experimental Methods and Results

We evaluate the computational complexity and performance of the OIG-HWO algorithm compared to other methods for approximation and replacement classifier tasks. In a replacement classifier, the OIG-HWO is used as a substitute in a ResNet18 He et al. (2016) style deep learning architecture and compares its testing performance to that of a scaled-CG (SCG) classifier Tyagi et al. (2021). We compare the performance of the proposed OIG-HWO algorithm with four other existing methods, namely, OWO-BP, OIG-BP, LM, and SCG. In SCG and LM, all weights are updated simultaneously in each iteration. However, in OWO-BP, OIG-BP, and OIG-HWO, we solve linear equations for the output weights and then update the input weights.

## 4.1 Experimental procedure

All the experiments are run on a machine equipped with a 3 MHz Intel i-7 CPU and 32 GB of RAM running the Windows 10 OS with PyTorch 1.13. We use the *k-fold* training with cross-validation and testing procedures to obtain the average training, validation, and testing errors. Each data set is split into $k$ non-overlapping parts of equal size ($k = 10$ in our simulations). Of this, ($k$-2) parts (roughly 80%) are used for training. Of the remaining two parts, one is used for validation, and the other is used for testing (roughly 10% each). The procedure is repeated till we have exhausted all $k$ combinations.

Validation is performed per training iteration (to prevent over-training), and the network with the minimum validation error is saved. After training, the saved weights and the testing data are used to compute a testing error, which measures the network's ability to generalize. At the end of the $k$-fold procedure, the average training and testing errors and the average number of cumulative multiplies required for training are computed. These quantities form the metrics for comparison and are subsequently used to generate the plots and compare performances.

In order to make a fair comparison of the various training methods for a given data set and fold, we use the same initial network for each algorithm, using net control Tyagi et al. (2022a). In net control, random initial input weights and hidden unit thresholds are scaled so each hidden unit's net function has the same mean and standard deviation. Specifically, the net function means are equal to .5, and the corresponding standard deviations are equal to 1. In addition, OWO is used to find the initial output weights. This ensures that all algorithms start at the same point in the weight space, eliminating any performance gains due to weight initialization. This is evident in all the plots, where we can see that all algorithms have the same starting MSE for the first training iteration. The final training error is hence not affected by different weight initializations. For each approximation datasets, we calculate the lowest MSE and probability of error, $Pe$ for classification datasets.

## 4.2 Computational Burden

One of the metrics chosen for comparison is the cumulative number of multiplies required for training using a particular algorithm. In this section, we identify the computational burden per training iteration for each of the algorithms compared.

Let $N_u = N + N_h + 1$ denote the number of weights connected to each output. The total number of weights in the network is denoted as $N_w = M(N + N_h + 1) + N_h(N + 1)$. The number of multiplies required to solve for output weights using OLS is $M_{ols}$, which is given by

$$M_{ols} = N_u(N_u + 1)\left[M + \frac{1}{6}N_u(2N_u + 1) + \frac{3}{2}\right] \tag{35}$$

The numbers of multiplies required per training iteration using BP, OWO-BP, OIG-HWO, and LM are given by

$$M_{bp} = N_v\left[MN_u + 2N_h(N + 1) + M(N + 6N_h + 4)\right] + N_w \tag{36}$$

$$M_{owo-bp} = N_v \left[ 2N_h(N+2) + M(N_u+1) + M(N+6N_h+4) + \frac{N_u(N_u+1)}{2} \right] + M_{ols} + N_h(N+1) \quad (37)$$

$$M_{oig} = M_{owo-bp} + N_v[(N+1)(3MN_h + MN + 2(M+N) + 3) - M(N+6N_h+4) - N_h(N+1)] + (N+1)^3 \quad (38)$$

$$M_{lm} = M_{bp} + N_v[MN_u(N_u + 3N_h(N+1)) + 4N_h^2(N+1)^2] + N_w^3 + N_w^2 \quad (39)$$

$$M_{scg} = 4N_v[N_h(N+1) + MN_u] + 10[N_h(N+1) + MN_u] \quad (40)$$

Note that $M_{oig}$ consists of $M_{owo-bp}$ plus the required multiplies for calculating optimal input gains. Similarly, $M_{lm}$ consists of $M_{bp}$ plus the required multiplies for calculating and inverting the Hessian matrix. $N_h^{\delta}$ is the number of new hidden units added at each growing step of the cascade correlation algorithm.

### 4.3 Approximation Dataset Results

We take mean square error (MSE) in the approximation datasets as the metric for various algorithm performances. In all data sets, the inputs have been normalized to be zero-mean and unit variance. This way, it becomes clear that OIG's improved results are not due to a simple, single-data normalization. Table 1 shows the specifications of the datasets used to evaluate the algorithm performances.

Table 1: Specification of approximation datasets

| Datasets | N | M | $N_v$ |
|----------|-----|-----|-------|
| Prognostics | 17 | 9 | 4745 |
| Remote Sensing | 16 | 3 | 5992 |
| Federal Reserve | 15 | 1 | 1049 |
| Housing | 16 | 1 | 22784 |
| Concrete | 8 | 1 | 1030 |
| White Wine | 11 | 1 | 4898 |
| Parkinson's | 16 | 2 | 5875 |

The number of hidden units for each data set is selected by first training a multi-layer perceptron with a large number of hidden units followed by a step-wise pruning with validation Tyagi et al. (2020). The least useful hidden unit is removed at each step until only one hidden unit is left. The number of hidden units corresponding to the smallest validation error is used for training on that data set. A maximum number of iterations is fixed for each algorithm, along with an early stopping criterion. The maximum training iterations for all algorithms are set to 1000.

For each dataset, we perform a 10-fold training with cross-validation and testing for the proposed OIG-HWO algorithm and compare with those listed in section 4, using the datasets listed in Table 1. Two plots are generated for each datasets. The average mean square error (MSE) for training from 10-fold training is plotted versus the number of iterations (shown on a $log_{10}$ scale), and the average training MSE from 10-fold training is plotted versus the cumulative number of multiplies (also shown on a $log_{10}$ scale) for each algorithm. Our results will present the average MSE achieved through training, along with the corresponding computational requirements. Given the constraints on the length of the paper, we have opted to display two plots for the initial dataset, while for the subsequent datasets, we depict the average MSE versus the cumulative number of multiplications. This approach provides a more compelling representation of both the learning and computational aspects of our study.

### 4.3.1 Prognostics Dataset

The Prognostics datasetile, called F-17 ipn (2022) consists of parameters that are available in the Bell Helicopter health usage monitoring system (HUMS), which performs flight load synthesis, which is a form of prognostics Manry et al. (2001). For this data file, 13 hidden units were used for all algorithms. In Figure 4, the average mean square error (MSE) for training from 10-fold validation is plotted versus the number of iterations for each algorithm. In Figure 5, the average training MSE from 10-fold validation is plotted versus the cumulative number of multiplies. From Figure 4, the overall training error for the proposed OIG-HWO overlaps with LM, with LM coming out on top by a narrow margin. However, the performance of LM comes with significantly higher computational demand, as shown in Figure 5. Prognostics data is a highly correlated/non-correlated dataset. The proposed OIG-HWO algorithm can give good performance despite these dependent features. The reason is that the input gains checks on the dependent features and reduces their effect while training. This aspect of input gains is not present in other comparing algorithms, and hence they are not as efficiently performing as the proposed algorithm. Another aspect of the features is that distinct distributions are essential in the algorithm's performance. This is proven by evaluating the algorithms using dependent features with and without. From Table 2, we observe that LM gives the marginally lowest mean squared error followed by OIG-HWO. It is worth emphasizing that the OIG-HWO algorithm achieves significantly lower testing errors than similar algorithms, except LM.

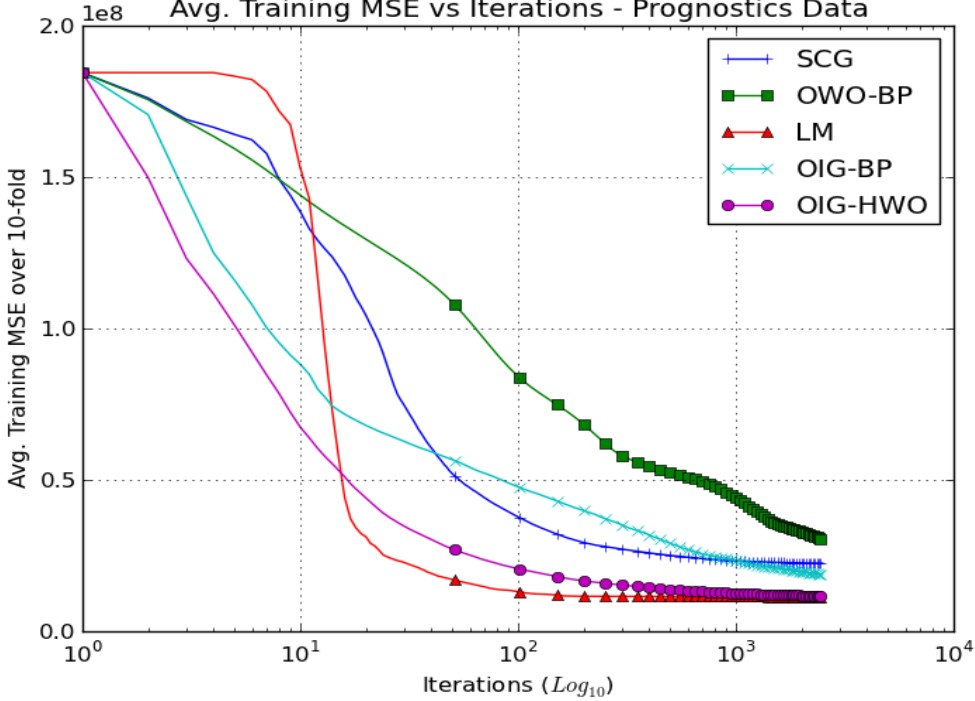

Figure 4: Average training MSE vs. Iterations for Prognostics data

### 4.3.2 Remote Sensing Dataset

The Remote Sensing data ipn (2022) represents the training set for inversion of surface permittivity, the normalized surface rams roughness, and the surface correlation length found in backscattering models from randomly rough dielectric surfaces Fung et al. (1992). For this dataset, 25 hidden units were used for all algorithms. From Figure 6, the average training MSE from the 10-fold procedure for OIG-HWO is better than all other algorithms being compared. Regarding computational cost, the proposed algorithms consume the least computation than OWO-BP, SCG, LM and OIG-BP. However, except OWO-BP, all algorithms utilize fewer computations about two orders of magnitude than LM. By assigning input gains, the proposed

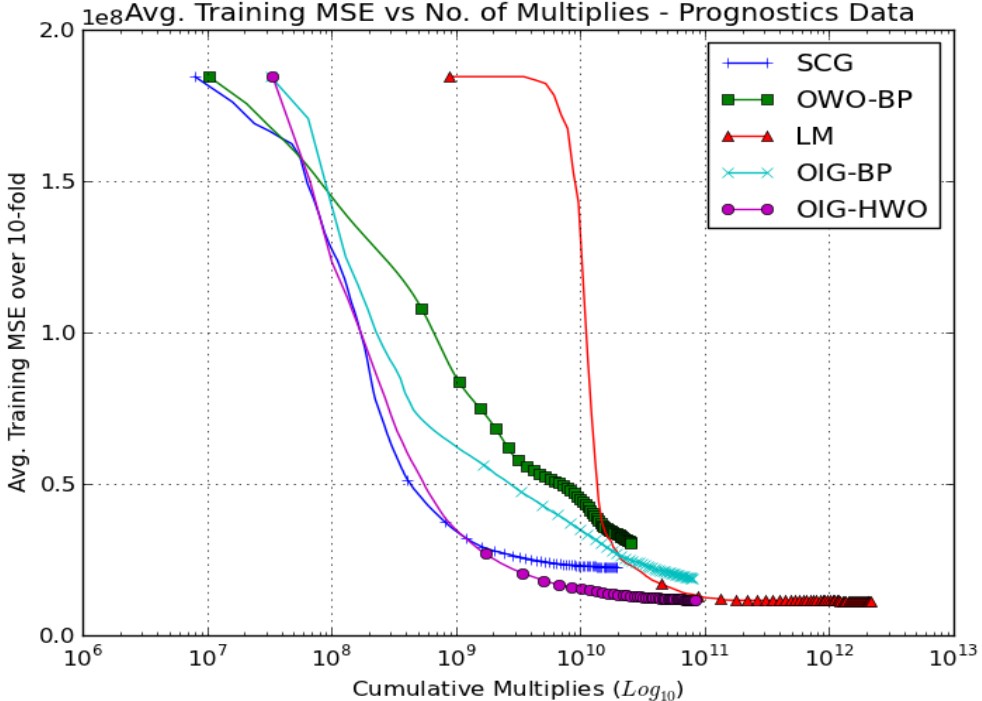

Figure 5: Average training MSE vs. cumulative multiplies for Prognostics data

OIG-HWO algorithm computes tailored weights for each input feature, reducing the effect of less important features and extracting useful information from the high dimensional dataset. From Table 2, we see that OIG-HWO has less testing error than all the other algorithms except LM. However, LM requires far more multiplier than OIG-HWO.

### 4.3.3 Federal Reserve Dataset

The Federal Reserve Economic Data Set fed (2011) contains economic data for the USA from 01/04/1980 to 02/04/2000 weekly. From the given features, the goal is to predict the 1-Month CD Rate similar to the US Census Bureau datasets us (2022). For this data file, called *TR* on its webpage, 34 hidden units were used for all algorithms. Figure 7 shows LM had the best overall training performance, with OIG-HWO a close. The proposed improvement to OIG performs better than OIG-BP, SCG, and OWO-BP without significant computational overhead. The proposed OIG-HWO algorithm can handle such data as it assigns high weights to features with acceptable variance than to features with very low variance. Results obtained from OIG-HWO act as a testament to the same. From Table 2, we observe that OIG-HWO has less testing error than the other algorithms.

### 4.3.4 Housing Dataset

The Housing dataset del (2011) is designed based on data provided by the US Census Bureau us (2022). These are all concerned with predicting the median price of houses in a region based on demographic composition and the state of the housing market in the region. House-16H data was used in our simulation, with 'H' standing for high difficulty. Tasks with high difficulty have had their attributes chosen to make the modeling more difficult due to higher variance or lower correlation of the inputs to the target. For this dataset, 30 hidden units were used for all algorithms. From Figure 8, the SCG algorithm at the end of training has less training error, followed closely by LM and OIG-HWO, respectively. The OIG-HWO algorithm adjusts the input gains and learns to assign lower weights to low variant features. Thus, the columns with almost

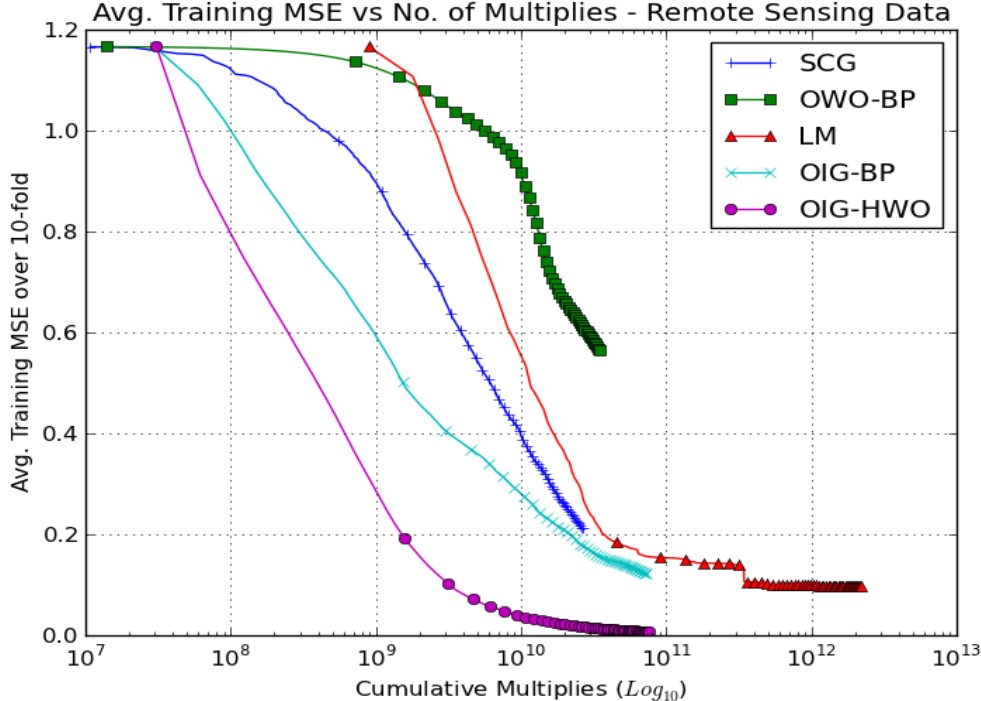

Figure 6: Average training MSE vs. cumulative multiplies for Remote Sensing dataset

constant values do not contribute toward the end goal. Table 2 shows the superiority of OIG-HWO having less testing error over other algorithms.

### 4.3.5 Concrete Compressive Strength Dataset

The Concrete dataset Yeh (1998)con (2013) is the actual concrete compressive strength for a given mixture under a specific age (days) determined by laboratory. The concrete compressive strength is a highly nonlinear function of concrete age and ingredients. For this dataset, we trained all algorithms with 13 hidden units. From Figure 9, LM has the best overall training error, followed by OIG-HWO and closely in third by SCG. Table 2, also supports the advantage from OIG-HWO with less testing error over other algorithms.

### 4.3.6 Wine data set

The Wine dataset Whi (2013) Cortez et al. (2009) is related to the wine variant of the Portuguese "Vinho Verde" wine. The inputs include objective tests (e.g., PH values), and the output is based on sensory data (median of at least three evaluations made by wine experts). Each expert graded the wine quality between 0 (very bad) and 10 (very excellent). For this dataset, we trained all algorithms with 24 hidden units. Figure 9 shows that LM has the best final training performance, followed very closely by OIG-HWO. OIG-HWO handles dependent features and accounts for the skewness in data, resulting in better results. Table 2 shows that OIG-HWO has substantially better testing performance than the other methods.

### 4.3.7 Parkinson's Dataset

The Parkinson's data set par (2013), Little et al. (2008) comprises a range of biomedical voice measurements from 42 people with early-stage Parkinson's disease recruited to a six-month trial of a telemonitoring device for remote symptom progression monitoring. The main aim of the dataset is to predict the motor and total UPDRS scores ($'motor'_{UPDRS}$ and $'total'_{UPDRS}$) from the 16 voice measures. For this data set, LM's performed better than OIG-HWO in terms of the training error, followed by the rest. However, LM requires

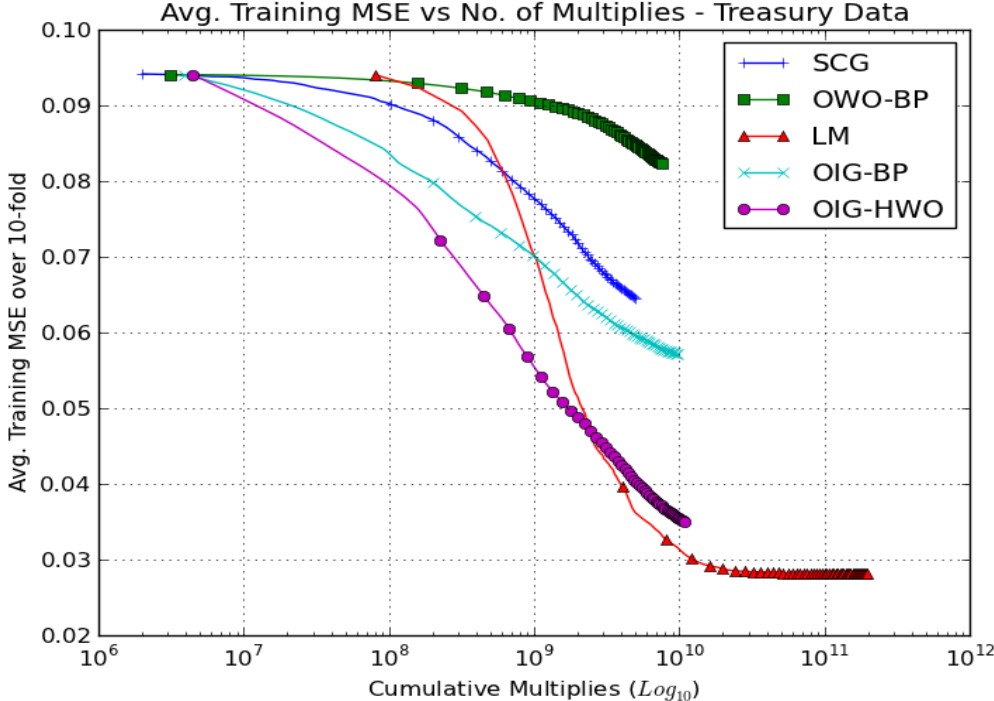

Figure 7: Average training MSE vs. cumulative multiplies for Federal Reserve dataset

a lot more computations to achieve the slight improvement, as evident in Figure 11. For this dataset, we trained all algorithms with 12 hidden units. From Table 2, we see that OIG-HWO has less testing error than other algorithms.

Table 2: 10-fold testing MSE results for approximation dataset, (best testing MSE is in bold)

| Dataset | OWO-BP | SCG | LM | OIG-BP | OIG-HWO |
|---|---|---|---|---|---|
| Prognostics | 3.2752E7 | 2.8224E7 | **1.4093E7** | 2.1490E7 | 1.4680E7 |
| Remote Sensing | 1.0655 | 0.8627 | 0.2954 | 0.7637 | **0.2094** |
| Treasury | 0.1391 | 0.3245 | 0.1072 | 0.1276 | **0.1036** |
| Housing | 17.2398E8 | 38.0949E8 | 11.9216E8 | 13.2538E8 | **11.7886E8** |
| Concrete | 30.5863 | 73.6012 | 27.7605 | 29.7421 | **27.1604** |
| White Wine | 0.5222 | 0.5812 | 0.4982 | 0.5027 | **0.4075** |
| Parkinson's | 131.1679 | 127.3441 | **123.2571** | 124.2698 | 123.57576 |

### 4.4 Discussion

From the training plots and Table 2, we deduce the following :

1. OIG-HWO is the top performer in 5 out of the 7 data sets in terms of testing MSE. The following best-performing algorithm is LM by a small margin. However, LM being a second-order method, its performance comes at a significant cost of computation – almost two orders of magnitude greater than the rest.

2. In terms of average training error, OWO-BP consistently appears in the last place on 4 of the 7 data sets, while SCG features in the last place on 3 out of 7 data sets. However, being first-order methods, they set the bar for the lowest computation cost.

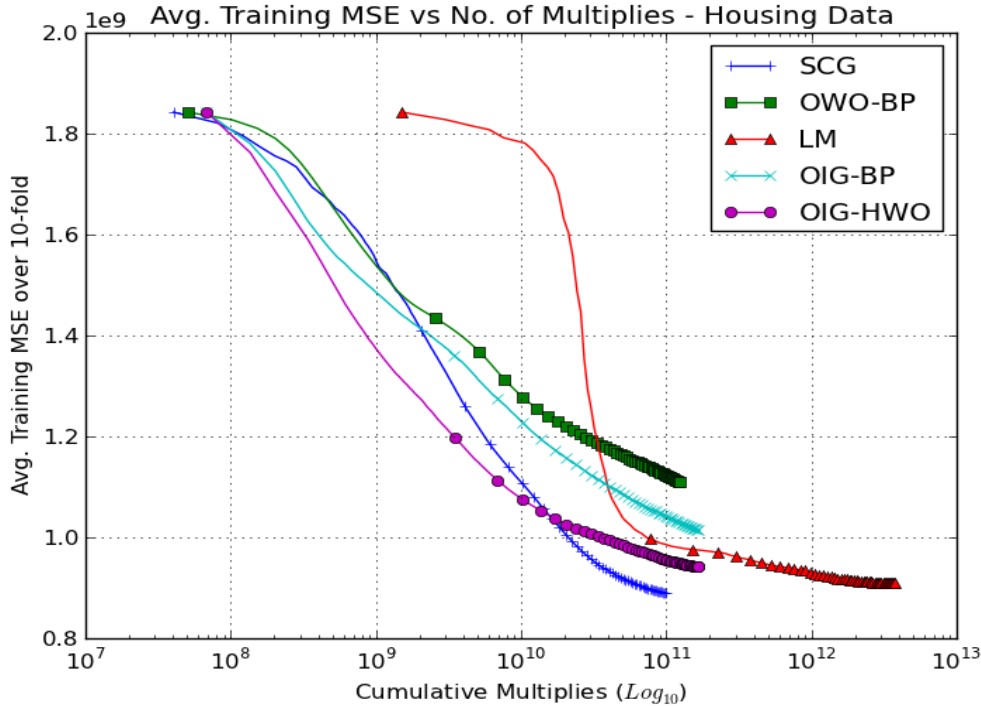

Figure 8: Average training MSE vs. cumulative multiplies for Housing Dataset

3. When SCG is potentially overtraining, OIG's training MSE is low for many of the initial iterations so early stopping will make training more efficient, even if SCG has less training error at the end.

4. Both OIG-BP and OIG-HWO always perform better than their predecessors, OWO-BP and OWO-HWO, respectively, on all data sets, and they are better than SCG. This performance is achieved with minimal computational overhead compared to SCG and OWO-BP, as evident in the training MSE plots vs. number of cumulative multiplications.

5. OIG-BP is never better than LM in training and testing MSE and consistently perfrom as the third best for all the datasets. OIG-HWO is always better than OIG-BP.

6. LM performs marginally better than OIG-HWO on two data set (*Prognostsics* and *Parkinson's* datasets) and has almost identical or worst ten fold testing MSE for the rest of the datasets.

7. Ignoring LM, which is a second order computationally heavy, overall, OIG-based algorithms (OIG-BP and OIG-HWO) are consistently in the top two performing algorithms.

As a general observation, both OIG-BP and OIG-HWO algorithms consistently outperform the OWO-BP algorithm in all three phases of learning, namely training, validation, and testing. The insertion of OIG into OWO-BP has been found to help enhance its performance. Furthermore, both OIG-BP and the improved OIG-HWO algorithms outperform SCG regarding the average minimum testing error. The OIG-HWO algorithm often performs comparably to LM but with minimal computational overhead. It is worth noting that while OIG-BP is an improvement over OWO-BP, it is not as effective as the OIG-HWO algorithm.

### 4.5 Replacement Classifier Datasets

We compare the OIG-HWO with CG-MLP and SCG using transfer learning. The classification datasets includes MNIST LeCun et al. (1998b), Scrap Kumar et al. (2022), Fashion-MNIST Kumar et al. (2022),

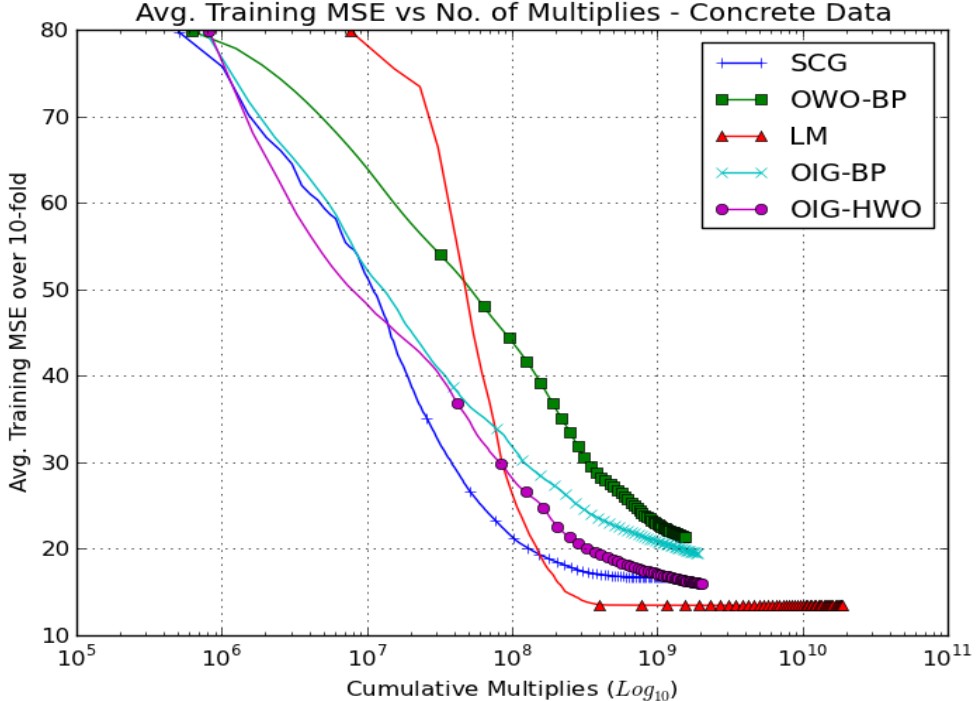

Figure 9: Average training MSE vs. cumulative multiplies for Concrete Dataset

CIFAR-10 Krizhevsky & Hinton (2009), SVHN Netzer et al. (2011), Cats-dogs Parkhi et al. (2012), Intel Image Int (2021). All the datasets are normalized to zero minimum and maximum one-pixel values. Table 3 shows the specifications of the datasets used to evaluate the algorithm performances. We studied a transfer learning comparison of the OIG-HWO, SCG, and OIG-BP algorithms using normalized classification datasets with pixel values ranging from zero minimum to one maximum. Table 3 outlines the specifications of the datasets used to evaluate algorithm performance.

Table 3: Replacement classifier datasets

| Datasets | N | M | $N_v$ | $N_v test$ |
|---|---|---|---|---|
| MNIST | 28 x 28 x 1 | 10 | 54000 | 6000 |
| Scrap | 28 x 28 x 1 | 2 | 14382 | 3595 |
| Fashion MNIST | 28 x 28 x 1 | 10 | 60000 | 10000 |
| CIFAR10 | 32 x 32 x 3 | 10 | 50000 | 10000 |
| SVHN | 32 x 32 x 3 | 10 | 73257 | 26032 |
| Cats and Dogs | 32 x 32 x 3 | 2 | 20000 | 5000 |
| Intel Image | 32 x 32 x 3 | 6 | 14034 | 3000 |

To create a replacement classifier in a deep learning architecture, we utilized the ResNet-18 architecture He et al. (2016) in the MATLAB 2021 Neural Network toolbox mat (2021). We trained ResNet-18 for each dataset, selecting the best validation accuracy ($P_e$) after a certain number of iterations. The training was performed using a learning rate of $1e-4$, 32 batch size, and Adams Kingma & Ba (2014) optimizer. We found the optimal $N_h$ value by a grid search for various $N_h$ values ($[5, 10, 15, 20, 30, 100]$). The feature vector extracted before this final layer was common to all datasets and contained 512 features. The best network was saved, and its final feature layer was extracted as input for each replacement classifier. ResNet-18 requires input images of size 224 x 224 x 3. We implemented an augmented image datastore pipeline with the option *colorprocessing=gray2rgb* to accommodate black and white images. We trained the network using a custom

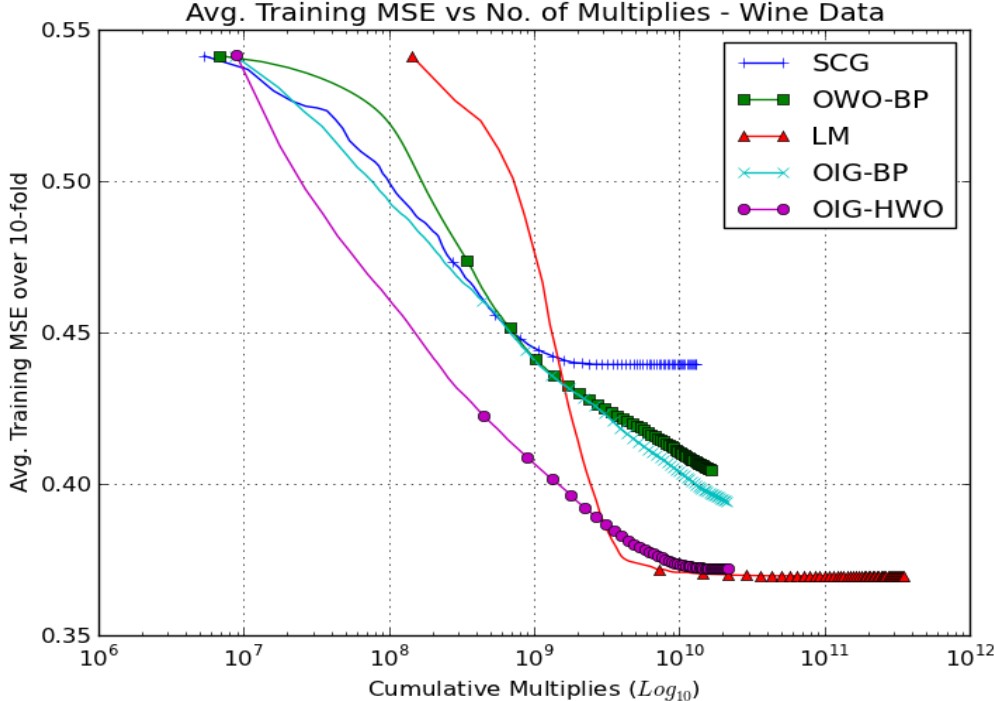

Figure 10: Average training MSE vs. cumulative multiplies for White Wine Dataset

final classification layer tailored to the specific number of classes for each dataset. After training, we replaced the final fully connected layer of ResNet-18, which has 1000 hidden units, with our replacement classifiers. Table 4 shows the superiority of OIG-HWO based classifier over other algorithms.

Table 4: 10 fold cross validation $P_e$ results for replacement classifier dataset, (best testing $P_e$ is in bold, for optimal $N_h$ values)

| Dataset | SCG/Nh | OIG-BP/Nh | OIG-HWO/Nh |
|---|---|---|---|
| MNIST | 0.39/30 | 0.37/20 | **0.368**/30 |
| Scrap | 0.728/20 | 0.554/10 | **0.509**/30 |
| Fashion MNIST | 5.705/100 | 5.812/5 | **5.366**/30 |
| CIFAR10 | 6.76/100 | 6.599/5 | **6.227**/100 |
| SVHN | 3.86025/30 | 3.632/30 | **3.619**/100 |
| Cats dogs | 4.516/100 | 4.504/100 | **4.32**/10 |
| Intel image | 9.483/100 | 9.287/10 | **9.02**/100 |

## 5 Conclusion and Future Work

In this study, we investigated the impact of linear input transformations on the training of MLPs. To do this, we developed the OIG-HWO algorithm, which optimizes the input gains or coefficients that scale the input data before the MLP processes it. The OIG-HWO algorithm uses Newton's method to minimize the error function with respect to input gains enhancing convergence of OIG-HWO. It has been shown that the learning behavior is different for functionally equivalent networks with different input transforms. It has also been shown that learning in the transformed network is equivalent to multiplying the original network's input

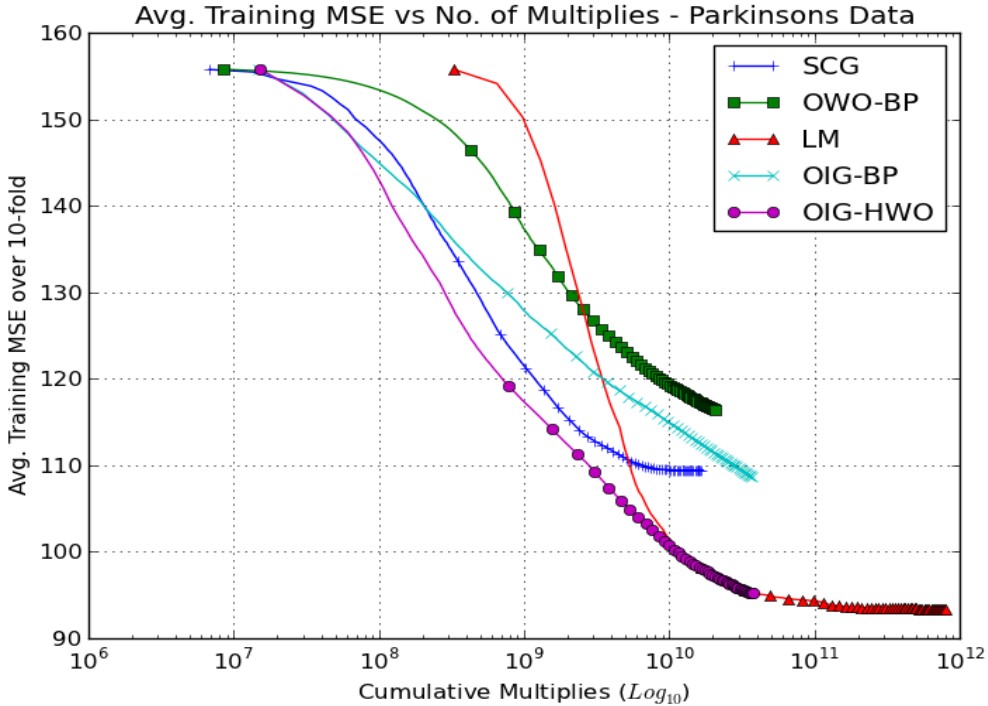

Figure 11: Average training MSE vs. cumulative multiplies for Parkinson's Dataset

weight gradient matrix by an autocorrelation matrix. It has also been shown that Newton's algorithm can be used to find the optimal diagonal autocorrelation matrix, resulting in the optimal input gain technique.

Beyond this, the OIG-HWO algorithm has some interesting characteristics desirable for deep learning architectures. Deep learning algorithms have performed exceptionally well in complex applications involving natural language, speech, images, and visual scenes. An underlying issue among these applications is the redundancy in data. Hence a typical pre-processing step in most deep learning applications is to apply whitening transformation to the raw data. HWO, as mentioned earlier is equivalent to back-propagation on whitened inputs. This means that OIG-HWO could serve as a building block for complex deep-learning architectures that could use the raw data directly without a pre-processing operation.

The OIG technique has been used to substantially speed up the convergence of OWO-BP, which is a two-stage first-order training algorithm. This algorithm was called OIG-BP. The OIG Gauss-Newton Hessian is a weighted average of the input weight Gauss-Newton Hessian, where the weights are elements of the negative input weight gradient matrix. OIG-BP was shown to be sub-optimal in the presence of linearly dependent inputs. Subsequently, OIG was applied to OWO-HWO to create an improved algorithm called OIG-HWO. Results from seven data sets showed that the OIG-based algorithms performed much better than two common first order algorithms with comparable complexity, namely SCG and OWO-BP. They come close to LM regarding the training error, but with orders of magnitude less computation. This is evident in all of the plots of training error versus the required number of multiplies and also from the expressions for the numbers of multiplies. Based on the results, we conclude that OIG-HWO is a strong candidate for shallow learning architectures and performs better than the SCG and OIG-BP algorithms as a replacement classifier.

For future work, the OIG technique can be extended to additional one and two-stage first-order algorithms, including standard BP, to other network types such as RBF networks and additional network parameters, yielding fast second-order methods rival LM's performance but with significantly reduced complexity.

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

## A  Appendix: Training weights by orthogonal least squares

OLS is used to solve for the output weights, pruning of hidden units Tyagi et al. (2020), input units Tyagi & Manry (2019) and deciding on the number of hidden units in a deep learner Tyagi (2018). OLS is a transformation of the set of basis vectors into a set of orthogonal basis vectors thereby measuring the individual contribution to the desired output energy from each basis vector.

In an autoencoder, we are mapping from an (N+1) dimensional augmented input vector to it's reconstruction in the output layer. The output weight matrix $\mathbf{W_{oh}} \in \Re^{N \times N_h}$ and $y_p$ in elements wise will be given as

$$y_p(i) = \sum_{n=1}^{N+1} w_{oh}(i,n) \cdot x_p(n) \tag{41}$$

To solve for the output weights by regression , we minimize the MSE as in (4). In order to achieve a superior numerical computation, we define the elements of auto correlation $\mathbf{R} \in \Re^{N_h \times N_h}$ and cross correlation matrix $\mathbf{C} \in \Re^{N_h \times M}$ as follows :

$$r(n,l) = \frac{1}{N_v} \sum_{p=1}^{N_v} O_p(n) \cdot O_p(l) \qquad c(n,i) = \frac{1}{N_v} \sum_{p=1}^{N_v} O_p(n) \cdot t_p(i) \tag{42}$$

Substituting the value of $y_p(i)$ in (4) we get,

$$E = \frac{1}{N_v} \sum_{p=1}^{N_v} \sum_{m=1}^{M} [t_p(m) - \sum_{k=1}^{N_h} w_o h(i,k) \cdot O_p(k)]^2 \tag{43}$$

Differentiating with respect to $\mathbf{W_{oh}}$ and using (42) we get

$$\frac{\partial E}{w_{oh}(m,l)} = -2[c(l,m) - \sum_{k=1}^{N_h+1} w_{oh}(m,k) r(k,l)] \tag{44}$$

Equating (44) to zero we obtain a $M$ set of $N_h + 1$ linear equations in $N_h + 1$ variables. In a compact form it can be written as

$$\mathbf{R} \cdot \mathbf{W}^T = \mathbf{C} \tag{45}$$

By using orthogonal least square, the solution for computation of weights in (45) will speed up. For convineance, let $N_u = N_h + 1$ and the basis functions be the hidden units output $\mathbf{O} \in \Re^{(N_h+1) \times 1}$ augmented with a bias of $\mathbf{1}$. For an unordered basis function $\mathbf{O}$ of dimension $N_u$ , the $m^{th}$ orthonormal basis function $\mathbf{O}'$ is defines as « add reference »

$$O'_m = \sum_{k=1}^{m} a_{mk} \cdot O_k \tag{46}$$

Here $a_{mk}$ are the elements of triangular matrix $\mathbf{A} \in \Re^{N_u \times N_u}$

For $m = 1$

$$O'_1 = a_{11} \cdot O_1 \quad a_{11} = \frac{1}{\|O\|} = \frac{1}{r(1,1)} \tag{47}$$

for $2 \leq m \leq N_u$, we first obtain

$$c_i = \sum_{q=1}^{i} a_{iq} \cdot r(q,m) \tag{48}$$

for $1 \leq i \leq m-1$. Second, we set $b_m = 1$ and get

$$b_{jk} = -\sum_{i=k}^{m=1} c_i \cdot a_{ik} \tag{49}$$

for $1 \leq k \leq m-1$. Lastly we get the coeffeicent $A_{mk}$ for the triangular matrix $\mathbf{A}$ as

$$a_{mk} = \frac{b_k}{[r(m,m) - \sum_{i=1}^{m-1} c_i^2]^2} \tag{50}$$

Once we have the orthonormal basis functions, the linear mapping weights in the orthonormal system can be found as

$$w'(i,m) = \sum_{k=1}^{m} a_{mk} c(i,k) \tag{51}$$

The orthonormal system's weights $\mathbf{W}'$ can be mapped back to the original system's weights $\mathbf{W}$ as

$$w(i,k) = \sum_{m=k}^{N_u} a_{mk} \cdot w'_o(i,m) \tag{52}$$

In an orthonormal system, the total training error can be written from (4) as

$$E = \sum_{i=1}^{M} \sum_{p=1}^{N_v} [\langle t_p(i), t_p(i) \rangle - \sum_{k=1}^{N_u} (w'(i,k))^2] \tag{53}$$

Orthogonal least square is equivalent of using the $\mathbf{QR}$ decomposition Golub & Van Loan (2012) and is useful when equation (45) is ill-conditioned meaning that the determinant of $\mathbf{R}$ is $\mathbf{0}$.

