# OpenReview forum: "Optimal Input Gain: All You Need to Supercharge a Feed-Forward Neural Network"
_TMLR — Rejected by TMLR_

### Review · Reviewer_zevo · 2023-10-12

**Summary Of Contributions:**

The paper combines existing optimization algorithms into one and tests the resulting one on several simple tasks and shallow neural networks.

**Audience:**

No

**Broader Impact Concerns:**

None.

**Claims And Evidence:**

No

**Requested Changes:**

1. The abstract should be fully re-written to match the paper and the claims.
2. Ideally writing should be improved.
3. There should be a comparison with standard first-order methods like SGD/Adam/AdamW/etc.
4. Sec. 4.2 would benefit a lot from the big-O notation for algorithm scaling.
5. The paper uses MNIST/CIFAR10 and some other datasets for a resnet18+adam. The could be used in a smaller network with the discussed algorithms too.

**Strengths And Weaknesses:**

**Potential issues**
> We establish in Malalur & Manry (2009) that input weights for MLP-1 and MLP-2 are related as

after Eq. 18. might be a breach of anonymity.


**Strengths**

The proposed algorithm seems to work better than some other existing ones on several tasks.

**Weaknesses**

1. Writing

My main issue with the paper is poor writing (but because of it I might be missing some other problems).
Overall writing is poor. I found many sentences hard to parse, but I cannot pinpoint exact problems.

The abstract specifically has very little to do with the actual paper. It talks about CNNs, but all experiments are done for MLPs (even the resnet18 experiment doesn't actually train a full resnet18 with the proposed algorithm, just the output linear layer). I can sort of see how the discussion about activation functions in the abstract follows from the proposed algorithm, but the main text doesn't explain it well (or at all?).

2. Abstract claims
> We show that these PWL activations work much better than ReLU activations in our networks for convolution neural networks and multilayer perceptrons. Result comparison in PyTorch for shallow and deep CNNs are given to further strengthen our case.

The paper really proposes a different optimization algorithm, not a different activation function (although you can view it as one). A fair comparison would be to take several architectures with ReLU/PWL trained with the same algorithm (even just SGD).

3. No SGD/Adam/AdamW/etc comparison

I don't understand why the method wasn't compared to the very standard first order methods.

4. Overall motivation of the work

We have very well-performing first-order algorithms for training neural networks. Since the proposed algorithm is not compared to those, I don't see the reason to use it over a computationally cheap SGD/Adam.

---

### Review · Reviewer_uxdC · 2023-10-13

**Summary Of Contributions:**

This paper studies the effects of linear input transformations on the training of Multi-Layer Perceptron (MLP) using the newly developed OIG-HWO algorithm. This algorithm focuses on optimizing the input gains or coefficients that pre-scale the input data for MLPs. The OIG-HWO algorithm emerges as a promising method in the realm of shallow learning architectures. It not only offers competitive performance metrics when compared with SCG and OIG-BP but also promises a reduction in computational demands. The authors have considered extensive testing across various domains and real-world scenarios in the experiments to compare the OIG-HWO algorithm with other novel methods.

**Audience:**

No

**Broader Impact Concerns:**

I don't think the submission requires a broader impact statement.

**Claims And Evidence:**

No

**Requested Changes:**

1. The citation format is not correct, especially in the introduction section. Please double-check and revise it.
2. The abstract needs to be revised. In the abstract, the authors mentioned the PWL activation function in the hidden layer, as their main contribution, which is not extensively explained in the main text. I think the authors need to rewrite this abstract.
3. The notations in section 2 are very ambiguous. For instance, we usually do not use $o_p(k)$ as a function since this is related to the order in probability notation and in theoretical computer science.
4. Typos between Lemma 1 and Lemma 2.
5. Where are the proofs of Lemmas 1-3?

**Strengths And Weaknesses:**

The authors base their methodology on established works (Malalur & Manry, 2009; Yu et al., 2004; Tyagi et al., 2022a), giving the study a solid foundation. The study not only introduces the OIG-HWO algorithm but also provides a thorough analysis of its structure, performance, and limitations. This paper has presented lots of experiments to compare the new algorithm's performance with existing algorithms and offer readers a contextual understanding of its advantages and effectiveness.

Full-batch gradient descent, mini-batch GD, and Adam are the most useful and common optimizers in deep learning. I do not know the motivation for introducing the OIG-HWO algorithm and how this new algorithm is related to common optimizers. Further experiments with different architectures, different dimensions, and scaling are needed to support this motivation.

---

### Review · Reviewer_GiGU · 2023-11-01

**Summary Of Contributions:**

This paper develops an optimisation method for a shallow neural network network, with only a single hidden layer. This results in three weight matrices, one connecting from inputs to a hiddden layer, W, another connecting from hidden to outputs, $W_{oh}$, and a linear skip-layer connection from inputs to outputs, $W_{oi}$. This method mainly focuses on optimising $W$, since the other two matrices are trivial to optimise by ordinary least squares, for a fixed $W$. This method first parametrises $W$ by $W‘.diag(d)$, where  each row of $W$ is scaled by an independent parameter. This vector effectively scales each dimension of the input. Then, we find the optimal d based on a second-order Newton's method to minimise the error. Since d is only a 1-d vector, the cost is smaller than a full second-order method on $W$. After finding $d$, $W'$ is updated with a slightly modified gradient descent. The authors conducted extensive experiments on multiple low-dimensional datasets, showing good performance and low computational cost of the proposed algorithms. In addition, the last experiment takes frozen ResNets trained on images, and slot in the shallow network in place of the last layer. The proposed algorithm again performs very well.

**Audience:**

No

**Broader Impact Concerns:**

Not very applicable here.

**Claims And Evidence:**

No

**Requested Changes:**

# Critical:
The authors should ideally address/fix all the points in the poor written quality part of this review. To address the significance issue, either of the following would be good to the reviewer:
1. quoting the original performance of the trained ResNet, and show that there is an improvement using the proposed method on the 1-hidden layer network.
2. Parametrise the weights of the ResNet by a scaling component, and optimise this by a second-order method just as the authors have done with d.

The authors should also provide satisfactory answers to ALL the points in the detialed comments EXCEPT c, d and e.

Given the amount of changes, I would give a strong rejection as it currently stands. I'd encourage the authors to consider re-submit it when the written quality is improved, becaues the technical idea could be of interest. The authors could benefit from taking a look at accpeted papers from TMLR and reformat the paper for this audience. e.g. go for a short submission and move redundant materials to the Appendix.

**Strengths And Weaknesses:**

# Upsides:
1. This paper is technically solid, focusing on a very well defined problem.
2. Although the written quality is of great concern (see below), I am able to understand the main point well and the method itself is reasonable and seems correct.
2. The experiments are very comprehensive, with sufficient details needed to replicate the results.
3. The experimental results are informative and shows useful metrics.


# Downsides:
## 1. Poor written quality.
This paper has all written issues one can enumerate. I do not criticise the technical quality of the paper, but the poor written quality pushes me to think that the authors are not careful enough and maybe even don't care about this paper.

1. All references are not enclosed in parenthesis. The authors could have never read the final submitted PDF.
2. There are gross misrepresentations of common practices in the deep learning community, such as saying line search is a common method for weight optimisation: it is not, the most common optimiser is still gradient descent or variants of it. This could be problem caused by different research fields, but in my opinion, readers of TMLR would not agree with these statements.
3. There are many overloaded uses of symbols, and the explanations do not distinguish two symbols that could mean the same thing but not entire certain.
4. The equations may be wrong to the extend that some details of the method is not comprehensible for a generic reader.
5. There are typos and unfilled placeholders in the Appendix.
6. There are definitely too much redundant details in the main paper, and some introductory contents are not clear but also not tightly related to the main contributions. I personally had great trouble understanding the first few pages, but then this did not seem to make it difficult to see the main points of the paper.

## 2. Very low significance.
Even though TMLR does not emphasise on impact, the paper focuses on such a tiny problem and setup that the chance of this getting adopted by any other researcher is almost zero. The authors need to either substantially expand the problem and network complexity, or demonstrate good performance on critical applications. Simply optimising a pre-trained ResNet would be useful, but it is unclear how much improvement there is by adopting this method.
1. The method as presented only applies to one set of weights in the most shallow network possible. It is possible to also parametrise all weights of more modern architectures, by parametrising the weights as done in this paper, and optimise the scaling part by second-order method. That way, the paper would be much more relevant to today's deep learning community.
2. Learning if full-batch. Can a minibatch version of this algorithm work? If not, again, this paper is less attractive to the deep learning community.
3. The ResNet experiment did not quote the original performance. Does adding the three-layer shallow network help? If so, then I would say more readers will start to care about this method.
4. The authors discussed the one benefit of the algorithm as not requiring whitening of the input data. This is such a tiny contribution, if any. Whitening is very simple to implement, if useful can be discovered by gradient methods.

## 3. Detailed comments

1. Can the authors clarify if the symbols A, C and R refer to the same mathematical objects throughout the paper? For example, do the C matrices in (11) and (33) mean the same thing?
2. Equation (12) is very confusing. The preceding text mentions that this is for the $p$'th data patter, but then the equation contains a sum of $p$. Is G simply the mean of the data repeated and tiled on top of each other? And why is this the correct gradient update? The error doesn't even show up in this equation.
3. What's the purpose of Section 3.1? I could not understand the purpose of this section, but later had no trouble understanding the later parts. In particular, why do we need to care about two networks that differ only by a linear transformation?
4. The Lemmas presented are either trivial or unclear for a reader to judge on the significance. Specifically, why is talking Lemma-3 important? This should seem obvious to any undergraduate student in a quantitative field. Also, for Lemma-2, in what sense are the two updates not equivalent? Why should we care when they are or are not equivalent?
5. Most of the detailed gradient expressions are unnecessary. (27)-(31) are simply different ways of writing down the Hessian, and I don't think they add any value for comprehension or implementation.
6. Section 4.2: I would love to believe these calculations are correct, but please give an information breakdown of each term.
7. There are huge numbers in the MSE reported in Table 2: on the order of 10^7? This does not sound right to me. If possible, please consider reporting the error divided by the variance of the ground-truth, so that the numbers are dimensionless.
8. Above eqn(46), please fix the reference placeholder.
9. Also, its "convenience", not "convineance" around the same place.
9. What's C in (32)? Can you give an example or such a matrix? If this is a main contribution of the paper, the reviewer would like to better understand it.

To put it simply, I get the main idea, it is reasonable, and the experiments seem okay as well. But the presentation has a lot of room for improvement that I cannot recommend for acceptance.

---

### Decision · Action_Editor_ECzf · 2023-12-11

**Recommendation:** Reject

**Comment:**

Main reasoning provided in section 'Claims and Evidence'. Summary: reviewers and AE criticise quality of writing and presentation of the paper (to a degree where the main claims are no longer supported by "accurate, convincing and clear evidence"), and (to a lesser degree) a lack of SGD as THE baseline to compare to and discuss. All reviewers suggest rejection; the authors did not respond to the reviews and did not upload a revised manuscript. Overall a clear reject for me.

**Audience:**

The main idea of the paper is interesting in principle (though limited in potential significance) and if executed well could be interesting to a small part of the TMLR audience.

**Claims And Evidence:**

The paper has a technically correct core idea with experiments that support the main idea. The core innovation is of very limited significance and fairly limited novelty, which is fine for TMLR. Unfortunately the current manuscript seems very rushed / unpolished. All reviewers criticise the lack of clarity and readability, have difficulties extracting the main claims, and point out significant mismatch between the abstract and discussion and the main body of the paper. Additionally (and as pointed out by the reviewers) the paper is missing important comparisons against standard SGD variants.

I agree with these two main issues raised by the reviewers - and do not think that the current manuscipt makes claims that are supported by "accurate, convincing and clear evidence". Not for technical reasons, but simply due to the unpolished writing (and the paper could use some trimming / re-prioritization of main content vs abstract). Though TMLR does not decide based on significance and impact, the quality of the writing and presentation and the scientific rigor do matter - TMLR is not an outlet for low-quality work. The current manuscript does not pass the bar. I believe the authors could address the issues of clarity and presentation, but since they chose not to respond to the reviews and upload a revised manuscript, the issues currently remain.

---

> ### Author Response · Authors · 2023-12-15
> **thank you for handling the paper**
>
> Thank you for giving the valuable comments from the reviewers and the Action editor. we sincerely appreciate it.
> We agree that the manuscript was a rush work and we are working to include all the comments and update the manuscript to submit at a later date.
>
> thank you again.